# Dynamically Tunable Assemblies of Superparamagnetic Nanoparticles Stabilized with Liquid Crystal-like Ligands in Organic Thin Films

**DOI:** 10.3390/nano13212908

**Published:** 2023-11-06

**Authors:** Zuzanna Z. Jańczuk, Agnieszka Jedrych, Sylwia Parzyszek, Anita Gardias, Jacek Szczytko, Michal Wojcik

**Affiliations:** 1Faculty of Chemistry, University of Warsaw, 1 Pasteur Street, 02-093 Warsaw, Poland; janczuk.zuzanna@gmail.com (Z.Z.J.); ajedrych@chem.uw.edu.pl (A.J.); sparzyszek@chem.uw.edu.pl (S.P.); 2Faculty of Physics, University of Warsaw, 5 Pasteur Street, 02-093 Warsaw, Poland; anita.gardias@fuw.edu.pl (A.G.); jacek.szczytko@fuw.edu.pl (J.S.)

**Keywords:** magnetic nanoparticles, iron oxide nanoparticles, self-assembly

## Abstract

The process of arranging magnetic nanoparticles (MNPs) into long-range structures that can be dynamically and reversibly controlled is challenging, although interesting for emerging spintronic applications. Here, we report composites of MNPs in excess of LC-like ligands as promising materials for MNP-based technologies. The organic part ensures the assembly of MNP into long-range ordered phases as well as precise and temperature-reversible control over the arrangement. The dynamic changes are fully reversible, which we confirm using X-ray diffraction (XRD). This methodology allows for the precise control of the nanomaterial’s structure in a thin film at different temperatures, translating to variable unit cell parameters. The composition of the materials (XPS, TGA), their structure (XRD), and magnetic properties (SQUID) were performed. Overall, this study confirms that LC-like materials provide the ability to dynamically control the magnetic nanoparticles in thin films, particularly the reversible control of their self-organization.

## 1. Introduction

Magnetic nanoparticles (MNPs) offer a wide range of opportunities for science and technology at the interfaces between biology, physics and chemistry [1,2,3]. Namely, numerous applications in cancer therapy [4,5], magnetic resonance imaging [6,7], sensing [8] and catalysis [9] are foreseen. Among various types of MNPs, iron oxide-based materials are particularly attractive due to their exceptional magnetic properties, and have been considered as potential candidates for constructing magnetic data storage devices [10,11]. To successfully implement this idea, large, well-organized, three-dimensional superlattices of monodispersed magnetic nanoparticles are required. Achieving such structures can be challenging when using the conventional top-down approach, due to the chemical instability of the surface of nanoscale components. Moreover, to fully capitalize on the potential of MNPs, such superlattices should be characterized with the precise positioning of MNPs with nanometer accuracy, and allow the control of their collective magnetic properties. Thus, after establishing the basic routes towards MNP superlattices the focus has shifted from understanding the self-assembly of individual components to dynamically manipulating long-range ordered, bulk-scale films of MNPs [12]. In this context, the use of electric or magnetic fields is promising for directing assembly; however, achieving reversible and controllable assembly of MNPs in a 3D-ordered solid-state lattices remains a challenge [13]. In other words, achieving intentional control of the movement and positioning of nano-objects, combined with the fact that magnetic fields can penetrate the volume of soft materials, enables the formation of ordered domains crucial for high-resolution technological application.

To fully understand the challenges of controlling assemblies of nanoparticles, it is convenient to categorize then into two types: those formed in solutions and in the condensed state [14,15]. By self-assembling nanoparticles in the presence of solvents, we can produce ensembles of nanoparticle aggregates and systems in a controlled and economical way [16,17]. A broad range of dynamic systems made of nanoparticles aggregates has been presented where assembly of nanoparticles aggregates under the influence of magnetic or electric fields [18], mechanical stress or light [19] offers a combination of speed and precision, as well as the ability to manipulate nanoparticle assemblies and aggregates [20,21,22]. Apart from systems containing azobenzene ligands, the remaining examples do not leave much room for post-factum, real-time dynamic manipulation of the material’s structure. Nevertheless, all the systems described to date have the same disadvantage: the concentration of dry-nanoparticle solids is relatively small, making their large-scale and industrial preparation cumbersome, which can markedly hamper their performance [23,24,25,26].

When analyzing the ways to obtain condensed state systems, the Langmuir–Blodgett (LB) method can be easily implemented for the large-area assembly of 2D ordered colloidal NPs or NCs arrays on solid substrates with very high precision in controlling gap sizes between NPs or NCs. However, this approach is limited mainly to 2D systems, which are less critical from the point of view of dynamic assemblies of nanoparticles and their potential applications. In order to realize large-scale 3D self-assembly of functional (plasmonic, magnetic) nanoparticles, liquid crystal assisted assembly is a very versatile approach that can offer flexibility, controllability, scalability and simplicity, not easily possible using other self-assembly techniques [27]. Liquid crystal (LC) systems provide several advantageous structural and physicochemical properties, and the incorporation of nanoparticles into the LC matrix can be utilized in developing nanoscale smart technologies that respond to specific and weak external stimuli in a controlled manner [28,29]. The use of LC offers a practical and uncomplicated approach for regulating material parameters. Through the manipulation of the external magnetic field intensity, a broad range of phase shifts can be attained [30]. However, it is crucial to manage the intricate equilibrium between topologically assisted colloidal self-assembly of magnetic nanoparticles and their anisotropic molecular interactions to achieve tunable optical properties of a soft colloidal LC-hybrid material [31]. The soft matrix is a good environment for arranging magnetic materials in clusters made of nanoparticles [20] or matrixes made of lipid-based liquid crystals [32]. By the choosing of proper organic molecules on nanoparticles or microparticles surface, control of the subsequent surface interactions leads to long-range ordered structure, allowing for optical [33], thermal [34] and electrical switching of obtained condensed systems [35,36]. By their soft nature, liquid crystal molecules provide spontaneous molecular order and self-assembly into composites with nanoparticles and LC-decorated nanoparticles. Several exciting and non-trivial structures and materials were obtained by dispersion of nano- and microparticles in a liquid crystal, a nematic fluid [37,38,39] of orientationally ordered lamellar phases [40,41].

Detailed information about the composites of LC systems and nanoparticles or microparticles can be readily obtained from thoughtful reviews [42,43]. LC-decorated nanoparticles are different from standard nanoparticles covered with simple alkyl molecules that, in a condensed state, are usually modified with alkyl derivatives of amines, organophosphorus or organosulfur compounds. These ligands are essential from the point of view of self-organization since more complex structures are required to induce an anisotropic shape or dynamic response [44]. One of the most studied hybrid soft systems is based on gold nanoparticles. Nevertheless, there have also been reports of other types of nanoparticles coated with promesogenic ligands that tend to form assemblies that exhibit soft and thermo-switchable structures [45]. The organic coating is responsible for the surface stabilization of NC’s and is a constituent of soft corona affecting a pseud-anisotropic shape of spherical nanoparticles [46]. Interesting reports have been carried out on anisotropic magnetic structures, but these are still systems based on a large excess of organic matrix [47]. Our research group contributes to developing topics on condensed systems built of promesogenic ligands and various types of nanoparticle-based soft thin films [48]. Recently, intermediate systems between fused soft nanoparticle and matrix systems have also been developed as chiral hierarchical supramolecular structures [49,50]. So far, no condensed thin films made of magnetic nanoparticles with a well-defined and long-range ordered structure reversibly switchable by temperature have been reported in the literature.

Here, we reported the fabrication of thermo-switchable hybrid thin films of approx. 5 nm iron oxide nanoparticles (IONPs) grafted by liquid crystal-like (LC-like) carboxylic acids ligands. We showed that structural factors and material composition have an impact on the self-organization at the nanoscale and, finally, on the magnetic properties of condensed arrays. The description and discussion of our results are divided into two sections. The first section describes the formation, qualitative characteristics and morphological properties of spherical IONPs and their composites. In the second section, we focus on the assessment of the magnetic properties of the nanoparticle assemblies in response to processes such as the assembly and disassembly of condensed arrays made of IONPs and parameters such as size and interparticle spatial properties. The magnetic properties were characterized by using the Superconducting Quantum Interference Device (SQUID) Magnetometer.

## 2. Materials and Methods

Chemicals. Iron chloride hexahydrate (Aldrich, 98%), sodium oleate (TCI, 95%), oleic acid (TCI, >85.0%), 1-octadecene (Aldrich, 90%), iron(III) acetylacetonate (Aldrich, 97%), 1,2-hexadecanediol (Aldrich, technical grade, 90%), and phenyl ether (Acros, 99%) were used as received without any further purification. All reagents for organic synthesis were obtained from Sigma-Aldrich (St. Louis, MO, USA). The reaction products were purified by column chromatography using SiliCycle Silica Flash P60 (40–63 µm, 60 Å, Québec, QC, Canada) at an atmospheric pressure or by crystallization. Thin-layer chromatography was performed using a silica gel 60 Å F254 (Merck, Darmstadt, Germany) precoated aluminum substrate and visualized using iodine vapor and/or a UV lamp (254 nm). All solvents were obtained from Sigma-Aldrich.

Thermogravimetric analysis (TGA). TGA analysis was performed with a TA Q50 V20.13 (TA Instruments, New Castle, DE, USA) analyzer. The measurements were carried out in the 100–500 °C range with a 10 °C min^−1^ heating rate in a nitrogen atmosphere.

X-ray photoemission spectroscopy (XPS). X-ray photoemission spectroscopy (XPS) was performed using a PHI 5000 VersaProbe–Scanning ESCA Microprobe (ULVAC-PHI, Chigasaki, Japan). The spectrometer was equipped with a quartz crystal monochromator and double beam system of charge compression with the Al X-rays electron gun and low-energy ions from the argon gun. Samples were prepared by casting a toluene suspension of nanoparticles on a silicon wafer followed by evaporation at room temperature.

XRD Measurements. XRD measurements at small angles were performed with a Bruker Nanostar system (Cu K α radiation (1.5406 Å), a parallel beam formed by cross-coupled Goebel mirrors, and a 3-pinhole collimation system, VANTEC 2000 area z detector, Billerica, MA, USA). The temperature of the sample was controlled with a precision of 0.1 K. Samples were prepared as thin films on Kapton tape or silica wafer substrates.

Transmission Electron Microscopy. TEM measurements were performed using a high-resolution JEM 1400 microscope (JEOL, Tokyo, Japan) equipped with tomographic holder and high-resolution digital camera CCD MORADA G2 (EMSIS GmbH, Münster, Germany) at Nencki Institute of Experimental Biology of Polish Academy of Sciences.

Differential Scanning Calorimetry. Calorimetric studies were performed with the TA DSC Q200 microcalorimeter (TA Instruments, New Castle, DE, USA). Samples with mass of 3 mg were sealed in aluminum pans and kept in nitrogen atmosphere during the measurement and both heating and cooling scans with a rate of 5 K min^−1^ were applied.

Superconducting Quantum Interference Device (SQUID). Magnetic measurements were conducted using Quantum Design MPMS XL-7 with 7.0 T magnet and temperature range 1.5–400.0 K.

### Synthetic Procedures

Synthesis of iron oxide nanoparticles of Series A. IONPs were obtained according to the procedure described in the literature [51]. In a three-necked flask iron(III) acetate (2 mmol) was mixed in phenyl ether (20 mL) with 1,2-hexadecanediol (10 mmol), oleic acid (6 mmol), and oleylamine (6 mmol) under nitrogen and was heated to reflux for 30 min. After being cooled to room temperature, nanoparticles were precipitated by adding ethanol. The product was dissolved in hexane in the presence of oleic acid and oleylamine and reprecipitated with ethanol.

Synthesis of iron oxide nanoparticles of Series B. Nanocrystals were obtained according to the method described by Park et al. [52]. The iron-oleate complex was obtained according to a well-described literature method. 10.8 g of iron chloride (40 mmol, Aldrich, 98%) and 36.5 g of sodium oleate (120 mmol, TCI, 95%) was dissolved in a mixture solvent composed of 80 mL ethanol, 60 mL distilled water and 140 mL hexane. The resulting solution was heated to 70 °C and kept at that temperature for four hours. When the reaction was completed, the upper organic layer containing the iron–oleate complex was washed three times with 30 mL distilled water. Next, hexane was evaporated off, resulting in an iron–oleate complex. A total of 36 g (40 mmol) of the iron-oleate complex and 5.7 g of oleic acid (20 mmol, Aldrich, 90%) were dissolved in 200 g of 1-octadecene (Aldrich, 90%). The reaction mixture was heated to 320 °C and then kept at this temperature for 30 min. The resulting solution containing the nanocrystals was then cooled to room temperature, and 500 mL of ethanol was added to the solution to precipitate the nanocrystals. The nanocrystals were separated by centrifugation.

Introducing LC-like ligand L to the surface of nanoparticles (HA and HB). Nanoparticles (5 mg) dispersed in 5 mL of toluene were mixed with LC-like ligand (10 mg) for 3 days at room temperature. Then, nanocrystals were precisely purified from ligands excess by precipitation with ethanol. The purity of the sample was confirmed by TCL chromatography from highly concentrated solutions of nanoparticles.

Composites. In order to obtain EA1 or EB1 material, a total of 30 μL of 1.0 mg mL^−1^ dispersion of HA or HB IONPs in toluene was mixed with 30 μL of 1 mg mL^−1^ solution of compound L in toluene (or with 60 μL or 120 μL of compound L solution to obtain EA2/EB2 and EA3/EB3, respectively). Then, the mixture was sonicated, and 3 μL of the mixture was dropcasted onto a TEM grid. Next, the sample was placed onto a heating table and subject to heating/cooling cycle between 30 and 130 °C, with a cooling rate of 3 °C min^−1^ and heating rate of 20 °C min^−1^. The process does not require an inert atmosphere.

Thin film preparation. The following description exemplified the preparation of the composite materials (EA and EB) on Kapton tape as a substrate. The composite material was dropcasted in portions using an automated pipette on the substrate at an elevated temperature (80 °C) to facilitate evaporation. Next, the sample was placed onto a heating table and subject to heating/cooling cycle between 30 and 130 °C, with a cooling rate of 3 °C min^−1^ and heating rate of 20 °C min^−1^.

## 3. Results and Discussion

### 3.1. Primary Iron Oxide Nanoparticles

The IONPs were synthesized according to two well-established protocols from the literature for the high-temperature decomposition of organic iron precursors (Figure 1a). As a result, we obtained two series of IONPs named Series A and B (from Sun [51] and Park [52] methods, respectively). In both cases, the primary ligand stabilizing the surface of nanoparticles was oleic acid (OLA). By using the right proportions of acetylacetonate to oleic acid (1:3) and oleic acid to iron oleate (1:4), we could produce small nanoparticles with similar diameters. These results were confirmed through measurements with transmission electron microscopy (TEM) and X-ray diffraction (XRD) (as seen in Figure 1b–f).

Series of IONPs differ in degree of monodispersity of nanocrystals core: 5.5 ± 0.9 nm (a broader logistic distribution of the nanoparticle diameter with two main groups 4.5–5.0 and 5.5–6.0 nm can be indicated) and 5.1 ± 0.6 nm for Series A and B, respectively (Figure 1d,f). The average distances between oleate-capped nanoparticles cores in the thin film, measured by XRD, were 6.1 ± 0.6 nm (Series A) and 6.2 ± 0.5 nm (Series B) and seem to be almost the same, but the impact of each fraction of nanoparticles is different. The distance difference obtained for TEM and XRD measurements is a consequence of considering the length of intertwining oleic acid chains (approx. 1 nm) in the XRD measurements.

### 3.2. Synthesis of Promesogenic Ligand (L)

We also synthetized an LC-like ligand (L) using organic synthesis. This ligand has a long and flexible alkyl chain that ends with a carboxylic acid group (Figure 2a). In our previous publications, similar molecules terminated with the thiol group induced a long-range liquid crystal behavior and thermotropic polymorphism of small gold nanoparticles [46,53,54]. In order to ensure the stability of nanoparticle’s cores, we replace the anchoring groups—from thiol to carboxyl. The synthetic path of carboxylic acid precursors is described in the Appendix A (Appendix A).

### 3.3. Hybrid Iron Oxide Nanoparticles

Primary IONPs (Series A and Series B) have been subjected to ligand-exchange reaction. The ratio of nanoparticle mass to mass of incoming ligands was 1:2. Nanoparticles were mixed in toluene for 3 days and then precisely purified from ligands excess by precipitation, as confirmed by TCL chromatography from highly concentrated dispersions of nanoparticles. According to previous studies [1], the amount of free ligand does not exceed 1% by weight. The modification process resulted in two series of hybrid nanocrystals HA and HB for Series A and B of primary nanoparticles, respectively (Figure 2b and Appendix A). To determine the composition of the organic shell on the nanoparticles’ surface, X-ray photoelectron spectroscopy (XPS) analysis was conducted before and after the ligand exchange process (Figure 2c,d). Accordingly, to previously reported data, characteristic doublet from Fe 2p_3/2_ and Fe 2p_1/2_ core-level electrons with binding energy values around 710 eV and 724 eV was observed. In the case of hybrid nanocrystals, the peak of energy values at about 724 eV is visible. Both spectrums show the presence of C and O elements. On the HR XPS survey of primary nanocrystals, two signals corresponding to carbon atoms in the oleic acid aliphatic chain (284.8 eV) and in carboxylate moiety (288,4 eV) are observed [55]. The appearance of nitrogen signal, coming from the amide group of ligand L confirms the successful exchange of ligands. Chemical composition based on XPS results is presented in Appendix A. The influence of the composition change of organic coating on nanoparticles organization was investigated by small-angle X-ray diffraction (SAXRD) (Figure 2e,f). Firstly, we can note that XRD diffractograms of all these samples comprised two main broad peaks. The second peak for primary IONPs, Series A and Series B, are less clearly visible than for HA and HB samples (after the ligands exchange); however, qualitatively these diffractograms are very similar. The obvious difference between the diffractograms of samples before and after the ligand exchange is that in the latter case, the XRD peaks are shifted towards smaller angles. Namely, the shift corresponds to a change of periodicity from 6.2 nm to 6.6 nm for Series A and from 6.1 nm to 7.6 nm for Series B. This change can be associated with the increasing size of molecules attached to the surface of the nanocrystals. In previous works of our group, it was often possible to precisely determine the symmetry of nanoparticle arrangement in the solid state based on the XRD diffractograms, as well as validate the assigned group based on the calculation of the volume of the metallic core and organic coating layer, [56,57,58] or even by visualization of the formed crystallites using TEM tomography [58]. As will be discussed later, for a number of XRD diffractograms of the materials obtained in this work we were able to assign FCC or BCC symmetries with high probability. However, in the case of HA and HB the width of the XRD peaks suggests that the samples exhibit short-range ordered structure, which hereafter we will refer to as SRO. Thus, a definitive assignment of symmetry is troublesome, although in some cases even such broad reflections could be used to suggest the mode of particle packing with the nearest neighbors.

Here, let us consider the HB sample. We first tried fitting FCC or BCC symmetries to the experimental diffractogram, as these symmetries are characteristic of the packing of spherical objects. However, in both cases, a poor match between the experiment and modeling was observed (Appendix A). Otherwise, the main XRD signal can be interpreted as a mean distance between the centers of the inorganic cores forming a square or hexagonal lattice. In such a case, if the overall shape is approximated as a sphere, the single particle volume would be $\frac{4}{3}\pi \;3.8^{3}$ that is ~229 nm^3^ or $\frac{4}{3}\pi \;{\left(\frac{3.8\cdot 2}{\sqrt{3}}\right)}^{3}$ that is ~354 nm^3^. The other way to estimate the volume of a single HB particle (Appendix A), is by using the TGA results. It shows that the mass ratio of organic matter to metal oxide is ~3.5:6.5, which can be recalculated to the corresponding volume ratio ~ $\frac{3.5:0.8}{6.5:5.2}=\frac{4.4}{1.25}=\frac{3.5}{1}$. Given the size of the metal oxide has a radius of ~2.55 nm, the overall particle volume, together with the organic material is ~2.55 nm, the overall particle volume, together with the organic material is ~315 nm^3^, which is well within the range estimated based on the XRD results.

Overall, we can conclude that the unequivocal assignment of particle aggregate symmetry is not possible; possibly, it is a mix of various symmetries, however rough estimates provide a reasonably good match between the calculations and TGA-based results. Thus, to not be speculative, we will not define the symmetry. Using the TGA technique, we have determined the ratio of the masses of organic and inorganic constituents of the resulting materials and their thermal stability (Figure 2g). The weight loss in the sample of primary Series B nanoparticles is 13.65%, which refers to the loss of approx. 205 OLA molecules from the nanoparticle surface. The thermogram obtained for hybrid nanocrystal HB showed a weight loss of about 65%.

HB material did not exhibit any changes in the assembly structure, that is the SRO arrangement was stable during heating in the wide tested temperature range. Since surface modification with promesogenic ligands does not always induce pseud-liquid crystalline order [57], we decided to conduct additional experiments involving the addition of free ligands as a soft matrix (EB composites).

### 3.4. Composites

In order to create dynamic arrangements of HA and HB systems, nanoparticles were combined with specific amounts of ligand L. Table 1 provides a summary of the different combinations and descriptions of the samples, based on the excess mass ratios of the ligands used. By adding an additional amount of ligand L in ratios of 1:1 to 1:4, composite materials were formed with the hybrid nanoparticles HA and HB (Table 1).

Thermal effects associated with phase transitions were investigated with differential scanning calorimetry (DSC) (Figure 3a). The obtained thermogram accordingly with polarizing optical microscopy (POM) observations with crossed polarizers indicate that this compound melts directly to an isotropic liquid at 75 °C. Upon cooling, this compound did not exhibit any thermal effects different to crystallization. The low-temperature phase is birefringent under crossed polarizers. LC-like ligand L did not exhibit any mesophases. Above 75 °C we observed melting to the isotropic phase. The composites obtained by combining HA nanoparticles with an excess of ligand L (EA2 and EA3) were characterized by a lower melting point than the pure ligand.

Based on DSC studies, the pure hybrid nanoparticles suspended in the ligand matrix did not exhibit any thermal events under the measurements condition. However, in the case of EA3 composite, an additional small thermal effect before the melting point is observed. The thermogravimetric analysis (TGA) of the magnetite nanoparticles were performed over the temperature range of 100–500 °C in a nitrogen atmosphere. For all samples, the main weight losses occurred in two distinct regions: from 250 to 350 °C and from 370 to 500 °C (Figure 4). Additional analysis of thermogravimetric data has been presented in the Appendix A (Appendix A). The thermograms of modified nanoparticles or composites with ligand excess shows that the significant mass loss in organic-coated nanoparticles occurs between the range 250–460 °C, which is higher than that for the pure OLA coated nanoparticles (ended around 440 °C). This shift in the temperature could be due to the multilayered adsorption of promesogenic molecules of ligand L, requiring a higher temperature for the vaporization. For all other samples, the weight loss was significantly higher due to promesogenic molecules attached and unattached to the surface of nanoparticles.

XRD studies of hybrid nanoparticles (HA) introduced to the ligand matrix evidenced a shift of the main XRD signal with the increasing amount of unbounded ligand L—from 6.6 nm for HA to 7.4 nm, 8.8 nm and 9.7 nm for EA1, EA2 and EA3, respectively (Figure 3b). This result indicates that the insertion of promesogenic ligands between nanoparticles can be used for achieving the statistical manipulation of distances between nanoparticles, at low temperature, which is interesting from the point of view of the applicability of this system. The evolution of the XRD pattern during heating for the EA2 composite material revealed two facts. Firstly, the high thermal stability of the obtained hybrid nanoparticles as well as composites (Figure 3c), as well as the reconfigurability of the particle assembly. Regarding the latter, the evolution of the XRD pattern during heating revealed the sharpening of the main XRD peak near 110 °C, which is related to the softening of the matrix and better distribution of the L molecules around hybrid nanoparticles. The positions of these signals remain essentially unchanged. However, above 150 °C a small shift to wide angles and a decrease in the intensity of the main signal indicated that a change in parameters within the same structure can be observed.

The other composite materials (EA1, EA3) are characterized by similar thermal behavior and thermal stability (Appendix A). The increase in nanoparticle distances was confirmed by TEM studies (Figure 3d–f and Appendix A). For structural studies of EA1 sample, it can be clearly seen that the mean-distance between nanoparticles is much smaller than in the layer of ligands between two nanoparticles’ cores observed for EA2. In the image obtained for the EA3 sample, it is possible to observe much larger distances between the nanoparticles than in the previous two samples. As shown in Figure 3g, composite EA2 exhibits higher-order organization in the soft matrix in a constant magnetic field. Nevertheless, we cannot study this organization due to the limitations of the SAXRD technique, and more research will be devoted to this phenomenon in future.

In an analogous way, promesogenic ligand substituted nanoparticles HB in the free-ligand matrix were prepared and studied. XPS studies confirmed the data of the sample composition. The XPS survey shows the presence of C, O and N (Figure 5a). The lack of signal indicating the presence of iron in a sample may result from relatively shallow irradiation of the material by X-ray beam. 1D XRD diffractograms revealed the presence of additional signals, which are characteristic for a well-defined hexagonal structure (Figure 5b).

When studying EB composites, the main XRD signal’s position changed in comparison to the HB particles, with varying amounts of added free ligands (ratios from 1:1 to 4:1 was examined). Thus, we measured larger interparticle distances, which can be ascribed to free ligands incorporation in between the particles. There is a relatively small difference between EB samples having different levels of free ligand doping at low temperatures, which suggests that at this temperature, the excess of organic ligands could be partially separated from the particles (Table 2).

When increasing the temperature for EB composites, we noted that relatively narrow Bragg peaks appeared in the diffractograms, suggesting a transition from SRO to long-range ordered assemblies, which allowed us to repeat the procedure for fitting the diffractograms with symmetries common for spherical objects. This approach is reasonable given the tendency of the ligands to melt at higher temperatures, and thus adopt a shape dependent on the shape of the inorganic nanoparticle core. Numerous examples of nanoparticles showing this behavior were previously reported for nanoparticles coated with LC-like ligands [59], and even alkyl ligands [58]. In detail, for the EB1 sample at 155 °C, we were able to reproduce the experimental diffractogram using face-centered cubic (FCC) symmetry (but not BCC). A similar short-to-long range order transition was observed for samples EB2 and EB3, above the melting point of the promesogenic ligand (Appendix A). Other structural events can be observed above 210 °C, but this was not investigated.

Since the EB samples exhibited long-range ordered structures at elevated temperatures, we can use the unit cell volumes of the assigned symmetries (BCC and FCC) to calculate the volume of a single entity forming the unit cell. The entity in these cases means an inorganic nanoparticle core, together with organic ligands bound to surface and free ligands infiltrating the assembly. In the case of the FCC unit cell, it comprises four entities.

The obtained values are much higher than those calculated for the HB particle without the free ligands added. In this context, it is interesting to follow the evolution of the volumes for the case of the EB3 sample, for which FCC symmetry can be assigned across a wide range of temperatures. Namely, single-entity volume varies from 1304.5, through 1502.6, up to 1539.5 nm^3^ for 65, 110 and 150 °C, respectively. This can be interpreted as the growing entropic demand of the organic molecules when elevating the temperature, a phenomenon that was already observed for nanoparticles coated with LC-like ligands [44].

Based on the presented data, we can conclude that using an excess of ligands in combination with monodisperse iron oxide nanoparticles resulted in the softening of the system, improving packaging and a relative increase in the mobility of the nanoparticles locked in the soft matrix. XRD studies confirmed that the obtained materials were thermostable up to 200 °C (Appendix A).

Magnetic measurements of selected samples (Figure 6a–d and Appendix A) showed that nanoparticles coated with oleic acid molecules (Series A and Series B) are characterized by higher magnetization than the same nanoparticles modified with promesogenic ligands (HB) or doped in the organic matrix after modification (EA2 and EB2). These observations are also confirmed by comparison of behavior of the original and modified nanoparticle samples in which magnetic properties are strictly dependent on the mutual position of nanoparticles in relative to each other. Measurements at temperatures of 0–80 °C (Figure 6e) were aimed at detecting changes in magnetization associated with possible phase transitions of the liquid crystal or other thermal events related to structural reorganization. Unfortunately, these changes were too subtle to be observed on the apparatus used for magnetic measurements. The only possible trace of a phase transition (probably related to the melting of the ligand matrix in the sample with excess ligand EB2) can be observed in Figure 6f, occurring at around 60 °C. Interesting results show the study of the magnetization of samples as a function of temperature (Figure 6e). The presence of organic ligands makes a larger, better separation of the magnetic cores, which causes antiferromagnetic interactions, resulting in a decrease in the magnetization of the entire material and a change in blocking temperatures. This phenomenon is especially visible in a very small field (50 Oe), because in the larger field, the applied magnetism determines the arrangement of the particles, not their interaction between each other.

## 4. Conclusions

Through our research, we successfully obtained small and monodisperse nanoparticles with a size of 5–6 nm. We then utilized a promesogenic molecule, a derivative of the 2NC8 ligand with a terminal carboxyl group, to modify the materials. Through measurements using XRD, XPS, TGA and TEM, we were able to confirm the composition and structural characteristics of the resulting materials. We have shown that partial replacement of primary ligands with promesogenic ligands increases the distance between nanoparticles and increases the volume of their unit cell. We also obtained composites of surface-modified nanoparticles and free promesogenic ligands. The obtained ordering of nanoparticles showed packing characteristics for SRO structures and was characterized by an increase in the volume of the unit cell associated with the incorporation of a matrix of regular nanoparticle structures. Most importantly, the temperature XRD measurements showed phase transitions between the SRO and FCC structures. Changes in the parameters of the elementary cells within the phases were also demonstrated. The phase transition was accompanied by a significant change in the unit cell parameters and volumes, and all temperature-dependent structure changes were completely reversible. The studied materials form a promising alternative in the context of creating thin and condensed films made of magnetic nanoparticles, whose internal structure and magnetic interactions can be dynamically controlled by temperature and, in the future, possibly by other remote factors as well.

## Figures and Tables

**Figure 1 nanomaterials-13-02908-f001:**
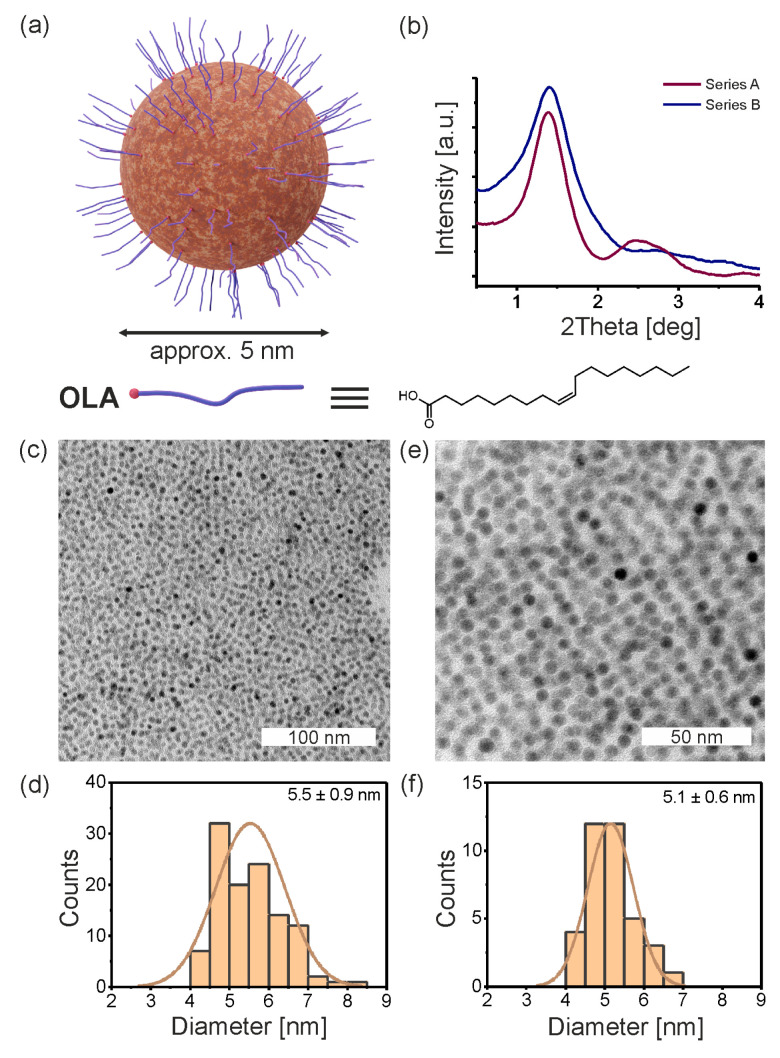
Primary IONPs. (**a**) The scheme of nanoparticle. Primary nanoparticles are obtained by thermal decomposition of the metal–oleate precursors in high boiling solvent and are grafted by oleic acid molecules. (**b**) 1D XRD diffractograms of IONPs of both series at 30 °C. The FWHM is smaller for Series B indicating higher correlation length which can be ascribed to lower dispersity of nanocrystals. (**c**,**d**,**f**) The TEM micrographs of IONPs of Series A and B, respectively with the histograms of size distribution presented in panel (**c**,**e**).

**Figure 2 nanomaterials-13-02908-f002:**
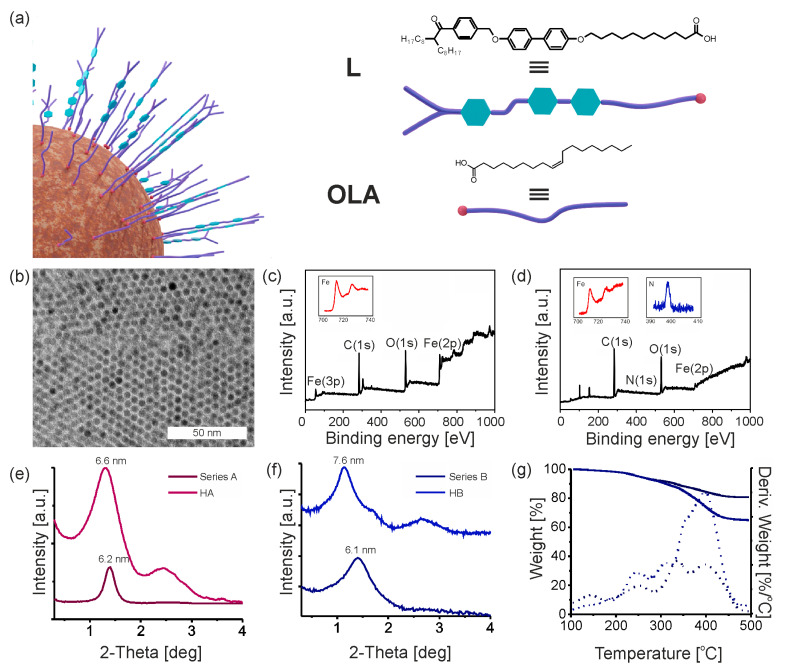
Hybrid nanoparticles HA and HB. (**a**) A scheme of the ligand-exchanged nanoparticles. Molecular structure of a promesogenic ligand L used for nanoparticle surface modification is presented. (**b**) The TEM image of thin film of hybrid nanoparticles after thermal annealing. (**c**,**d**) XPS survey spectra of primary and hybrid nanoparticles (HB), respectively. HR-XPS spectra of relevant elements are depicted in the insets. (**e**,**f**) 1D XRD diffractograms collected at 30 °C for IONPs of both series before and after the ligand-exchange process. (**g**) Thermogravimetric analysis (TGA) of primary and hybrid nanoparticles.

**Figure 3 nanomaterials-13-02908-f003:**
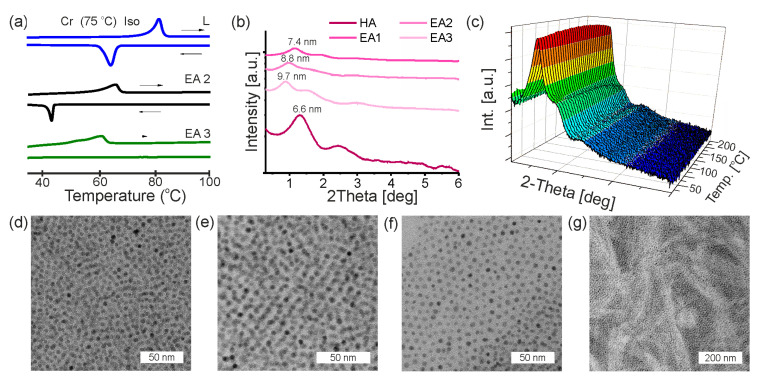
Composites with HA nanoparticles. (**a**) DSC thermograms of ligand L and two composite materials; (**b**) the comparison of 1D XRD diffractograms for pure HA nanoparticles and composites based on this type of IONPS at 110 °C; (**c**) temperature evolution of the XRD diffractogram of EA2 composite; (**d**–**f**) TEM photos for thin layers of EA1, EA2 and EA3 samples, respectively; (**g**) TEM picture of condensed sample of EA2 dropcasted on TEM grid in constant magnetic field.

**Figure 4 nanomaterials-13-02908-f004:**
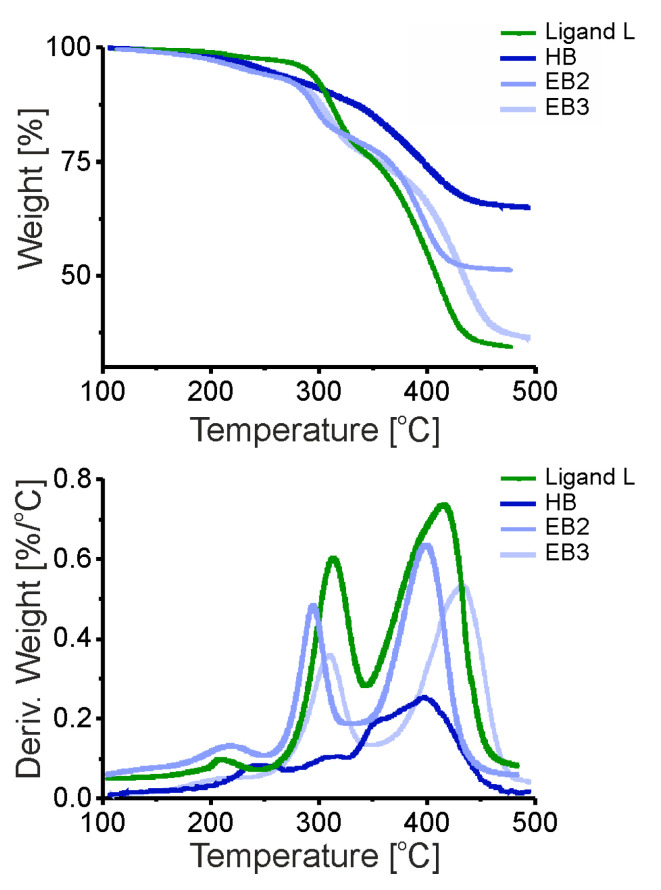
Thermograms of composites based on HB nanocrystals. Ligand L is presented for reference.

**Figure 5 nanomaterials-13-02908-f005:**
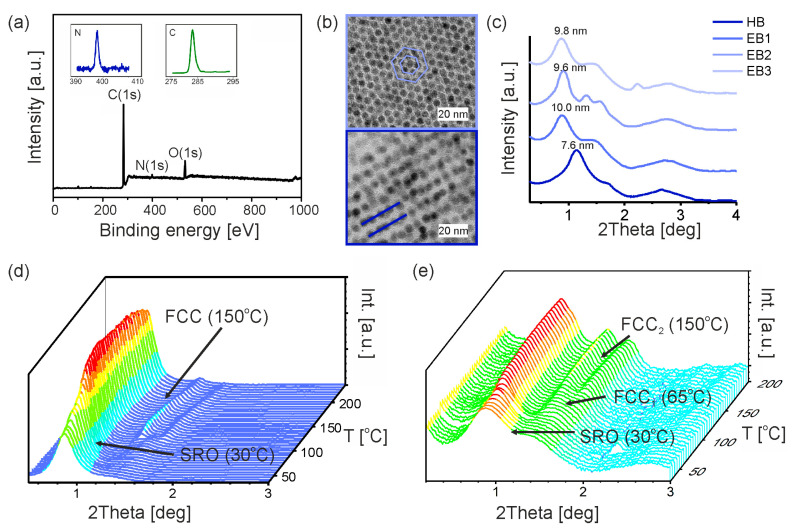
Composites with HB nanoparticles. (**a**) XPS survey for EB1 material; (**b**) TEM images of HB and EA3 samples after thermal annealing; (**c**) X-ray patterns for HB, EB1, EB2 and EB3 samples at 70 °C showing the shift of the main X-ray signal towards the smaller angles along with addition of the matrix; (**d**,**e**) temperature evolution of the scattering signal for EB1 and EB3.

**Figure 6 nanomaterials-13-02908-f006:**
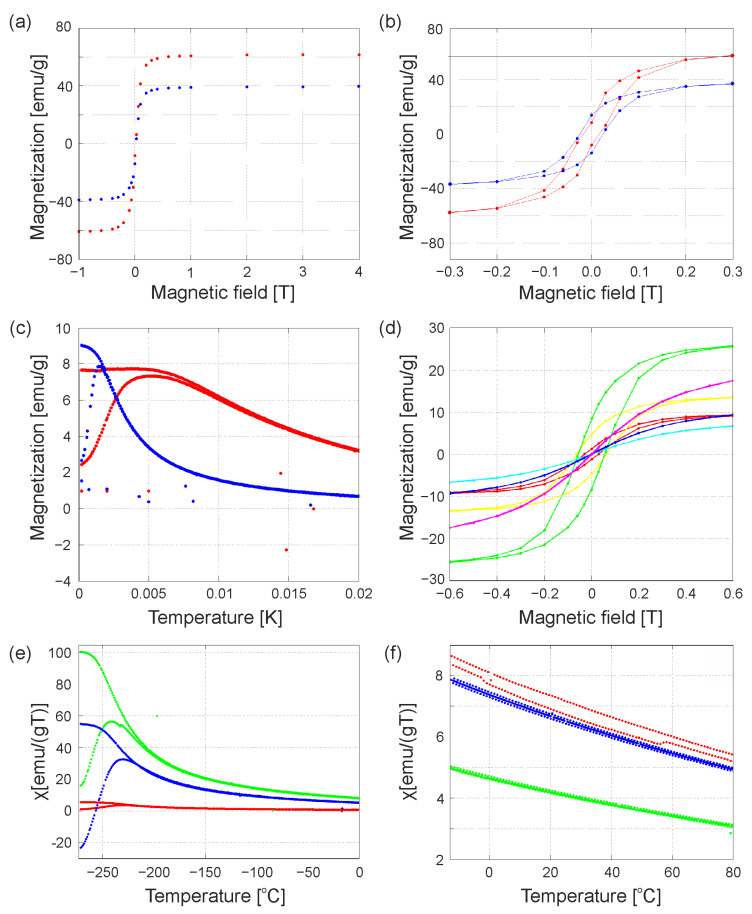
Magnetization measurements. (**a**) Magnetization vs. applied magnetic field at 2 K for samples Series A (red) and EA2 (blue); (**b**) magnified range regarding magnetization hysteresis for Series A (red) and EA2 (blue) samples at T = 2 K; (**c**) magnetization of samples as a function of temperature in the 0.005 T field for Series A (red) and EA2 (blue); (**d**) graph of sample magnetization versus magnetic field for Series B (in 2 K red, in 300 K light blue), EB2 in 2 K yellow, in 300 K blue), HB (in 2 K green, in 300 K magenta); (**e**) graph of materials magnetization vs. temperature for samples: Series B (red), EB2 (green) and HB (blue) in ZFCFC 50 Oe mode; (**f**) magnetization of EB2 in different modes: ZFCFC 50 Oe (red), 50 Oe (green) and 500 Oe (blue).

**Table 1 nanomaterials-13-02908-t001:** Summary of the obtained hybrid materials.

Primary IONPs	Hybrid IONPs	Free Ligand to IONPs Ratio (by Weight)	Composites with Ligand Excess
Serie A	HA1	1:1	EA1
		2:1	EA2
		4:1	EA3
Serie B	HB1	1:1	EB1
		2:1	EB2
		4:1	EB3

**Table 2 nanomaterials-13-02908-t002:** Detailed description of nanoparticle (Series B) assemblies.

	Space Group		Peaks Positions [nm]	Cell Parameters [nm]	Cell Volume [nm^3^]	Cell Volume per Nps [nm^3^]
EB1	SRO (30 °C)	P63/mmc				
	FCC (150 °C)	Fm-3 m	(111) − 10.3 (002) − 8.9 (022) − 6.3 (311) − 5.4 (222) − 5.1	a = 17.8	5709.0	1427.3
EB2	SRO (30 °C)	P63/mmc				
	SRO (110 °C)	P63/mmc				
	FCC (220 °C)	Fm-3 m	(111) − 9.9 (002) − 8.6 (022) − 6.1 (311) − 5.2 (222) − 5.0	a = 17.2	5047.4	1261.9
EB3	SRO (30 °C)	P63/mmc				
	FCC_1_ (65 °C)	Fm-3 m	(111) − 10.0 (002) − 8.7 (022) − 6.1 (311) − 5.2 (222) − 5.0	a = 17.3	5218.0	1304.5
	FCC_2_ (110 °C)	Fm-3 m	(111) − 10.5 (002) − 9.1 (022) − 6.4 (311) − 5.5 (222) − 5.3	a = 18.2	6010.4	1502.6
	FCC_3_ (150 °C)	Fm-3 m	(111) − 10.6 (002) − 9.2 (022) − 6.5 (311) − 5.5 (222) − 5.3	a = 18.3	6157.8	1539.5

## Data Availability

Not applicable.

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
