# Peer review of "Dynamically Tunable Assemblies of Superparamagnetic Nanoparticles Stabilized with Liquid Crystal-like Ligands in Organic Thin Films"

_nanomaterials, 2023, doi:10.3390/nano13212908_

Round 1

Reviewer 1 Report

Comments and Suggestions for Authors

This manuscript reports the synthesis, functionalization and self-assembly of small (5nm diameter) iron oxide nanoparticles (NP) in a mesogenic matrix. The experimental part is very thorough and well performed, but the data analysis and the ensuing conclusions are deficient, so that the main conclusions (and even some terms in the title) are not well substantiated. More details are given below, but overall major revision would be needed before publication.  

1. As a general observation, the SAXRD data is not clearly presented: the curves overlap, so the details are difficult to see, e.g. in Figs. 3b, 3c and 5c. Shifting the individual spectra vertically would render the display much more legible. It would also be useful to mark the Bragg peak positions.
2. The core material is identified in passing as magnetite, with no further details: I suppose this conclusion is based on the magnetic measurements, but that data is presented without discussion. At a minimum, the coercivity should be indicated for each hysteresis loop and compared with literature data. Are the particles truly superparamagnetic?
3. The ligand addition protocol and the TGA measurements would be enough to provide the NP volume fraction, which should be compared with the structural models (which yield the same parameter). This is important, since the samples might not be homogeneous: chains and aggregates are often present in this type of systems.
4. There might be some confusion, since the HCP phase has a constant ratio c/a=1.633, unless the values given in Table 2. Are the authors using the standard notations?
5. More generally, how were the phases identified? How many peaks are used, and what is their position ratio? No such information is provided, either in Table 2 or in the figures. Some spectra only exhibit one or two peaks, so I'm skeptical about these claims. This is one of the major shortcomings of the manuscript.
6. The phrase "liquid crystal-like" in the title is unclear: the particles themselves are supposed to form a 3D crystalline structure, while the matrix mesogens do not actually form an LC phase. I suggest the authors drop the qualifier.
7. The second major weakness is the reversibility claim, which is made both in the abstract and conclusion, but without any proof. This is one of the main points of the manuscript, so it requires development.

Minor points:
8. Some grammar errors and awkward phrases are present, mainly in the introduction.
9. I do not understand what the "bilayer" is on page 10.
10. The reasoning on page 11 after Table 2, explaining which lattice parameters do and do not change with L addition, is not clear, especially insofar it invokes "the smaller volumes of free spaces between nanoparticles in the condensed state". Where would this free space exist?
11. The "significant change" in unit cell volume (page 13) is presumably related to the different number of particles per unit cell in the two lattices; the volume per particle would be a more relevant parameter.
12. The magnetic field effect on the NP organization is not studied beyond showing the image in Fig. 3g.

Comments on the Quality of English Language

Some grammar errors and awkward phrases are present, mainly in the introduction.

Reviewer 2 Report

Comments and Suggestions for Authors

This paper describes the preparation and characterization of magnetic nanoparticles and their composites with the excess LC-like ligand. Although this reviewer appreciates the authors’ efforts, the manuscript is below the acceptable level of a research paper. In short, descriptions are too vague in contrast to their seeming fluency in writing. There are many problems.

1. In the lengthy Introduction, the achievement to be reported is unclear.

2. Unless the SAXD results are indicated against the wavenumber, the wavelength of the used X-ray should be explicitly specified.

3. l.236, It is unclear how XRD results are used for confirmation. Fig. S1 seems to be the same as Fig. 1df.

4. There is no description of the preparation of the film announced in Introduction.

5. Throughout the manuscript, the explanation of experimental results is insufficient.

a) The caption to Fig. 1b does not hold without the peak assignments.

b) No evidence is shown for the statement in l. 271-273. Explain Fig. 2e and f. 

c) Indicate the indices in Fig. 5c.

d) l. 385-387 Explain in detail in terms of reflection indices.

6. Fig. 2 c and d may be combined for a better presentation.

7. Using HCP is problematic when considering the change in the cell constant. (FCC is acceptable because of the absence of “close” in its name.

8. l.278 It is hard to understand the logic of using “However.”

9. l.304 Fig. 3a is insufficient to claim the absence of a mesophase.

10. l.306 The thermogram is not a name of an analytical technique.

11. The presentation of the mass loss relative to the total mass (Fig. 4a) is inadequate for this purpose. It is better to present them relative to the total mass of L in the sample for better comparison.

12. l.346, 347 What is the bilayer?

13. The statement in l.375-376 seems impossible unless the system is phase-separated.

14. The abscissa of Fig. 5c seems incorrect. What are the scattered points in this figure?

15. l.415 What is the “prediction”?

16. l.444 The change in the cell parameter comes from the lattice setting. The statement is nonsense.

17. Minor points

a) The fifth author lacks the number indicating the affiliation.

b) Some abbreviations, such as SAXRD and SQID, are non-standard. 

c) Inconsistency exists in naming samples: Series vs Serie.

d) l.355 very high temperature stability -> very high thermal stability or very high temperature-stability

Comments on the Quality of English Language

The authors should be careful for choosing wording.

Round 2

Reviewer 1 Report

Comments and Suggestions for Authors

The authors have thoroughly reviewed the manuscript and have addressed all the points I had raised. I therefore recommend publication as is.

Reviewer 2 Report

Comments and Suggestions for Authors

The authors revised the manuscript in the right direction. However, this reviewer thinks it is insufficient.

1.     Since the subject of this paper is the preparation and characterization of thin films, the description of the preparation, though briefly added, is entirely insufficient. What are the atmospheres for drop-casting and subsequent heat treatment? What is the thickness of the film? Does it affect the reported results? Is the word “thin” essential? In other words, are the results different for thick films?

2.     Simple substitutions of HCP with SRO do not solve the problem. The explanation of what the authors judged from the diffractions and considered is missing in the text. Without such explanations, readers cannot understand the story. Further, the peak assignment assuming the HCP-like packing is necessary to calculate the volume per NP. Note that the unit cell volume per particle is nonsense. The number of particles is fixed for lattice structures.

3.     Minor points:

a)     Some words are missing between lines 128 and 129.

b)     “Serie” is not a standard English word.

c)     SRO is not defined in the text.

d)     There are some typos.
